# Introduction to the EQIPD quality system

Anton Bespalov[1†]*, René Bernard[2,3,4†], Anja Gilis[5†], Björn Gerlach[1†], Javier Guillén[6], Vincent Castagné[7], Isabel A Lefevre[8], Fiona Ducrey[9], Lee Monk[10], Sandrine Bongiovanni[11], Bruce Altevogt[12], María Arroyo-Araujo[13], Lior Bikovski[14,15], Natasja de Bruin[16], Esmeralda Castaños-Vélez[2], Alexander Dityatev[17,18,19], Christoph H Emmerich[1], Raafat Fares[20‡], Chantelle Ferland-Beckham[21], Christelle Froger-Colléaux[7], Valerie Gailus-Durner[22], Sabine M Hölter[23], Martine CJ Hofmann[16], Patricia Kabitzke[24,25], Martien JH Kas[13], Claudia Kurreck[2], Paul Moser[26,27], Malgorzata Pietraszek[1], Piotr Popik[28], Heidrun Potschka[29], Ernesto Prado Montes de Oca[30,31,32], Leonardo Restivo[33], Gernot Riedel[34], Merel Ritskes-Hoitinga[35,36], Janko Samardzic[37], Michael Schunn[38], Claudia Stöger[22], Vootele Voikar[39], Jan Vollert[40], Kimberley E Wever[35], Kathleen Wuyts[41], Malcolm R MacLeod[42], Ulrich Dirnagl[2,3,4], Thomas Steckler[5]

[1]PAASP, Heidelberg, Germany; [2]Department of Experimental Neurology, Charité Universitätsmedizin, Berlin, Germany; [3]NeuroCure Cluster of Excellence, Charité - Universitätsmedizin Berlin, corporate member of Freie Universität Berlin, Humboldt-Universität zu Berlin, and Berlin Institute of Health, Berlin, Germany; [4]QUEST Center for Transforming Biomedical Research, Berlin Institute of Health at Charite, Berlin, Germany; [5]Janssen Pharmaceutica NV, Beerse, Belgium; [6]AAALAC International, Pamplona, Spain; [7]Porsolt, Le Genest-Saint-Isle, France; [8]Rare and Neurologic Diseases Research, Sanofi, Chilly-Mazarin, France; [9]Integrity and Global Research Practices, Sanofi, Chilly-Mazarin, France; [10]Research and Clinical Development Quality, UCB, Slough, United Kingdom; [11]Quality Assurance, Novartis Institutes for BioMedical Research, Novartis Pharma, Basel, Switzerland; [12]Pfizer Inc., Silver Spring, United States; [13]Groningen Institute for Evolutionary Life Sciences, University of Groningen, Groningen, Netherlands; [14]School of Behavioral Sciences, Netanya Academic College, Netanya, Israel; [15]The Myers Neuro-Behavioral Core Facility, Sackler School of Medicine, Tel Aviv University, Tel Aviv, Israel; [16]Fraunhofer Institute for Translational Medicine and Pharmacology ITMP, Frankfurt am Main, Germany; [17]Molecular Neuroplasticity, German Center for Neurodegenerative Diseases, Magdeburg, Germany; [18]Center for Behavioral Brain Sciences, Magdeburg, Germany; [19]Medical Faculty, Otto-von-Guericke University, Magdeburg, Germany; [20]Charles River Laboratories, Safety Assessment, Lyon, France; [21]Cohen Veterans Bioscience, Boston, United States; [22]German Mouse Clinic, Institute of Experimental Genetics, Helmholtz Zentrum München, German Research Center for Environmental Health, Neuherberg, Germany; [23]Institute of Developmental Genetics, Helmholtz Zentrum München, German Research Center for Environmental Health, and Technical University Munich, Munich, Germany; [24]PAASP US, Ridgefield, United States; [25]The Stanley Center for Psychiatric Research, Broad Institute of MIT and Harvard, Cambridge, United States; [26]Cerbascience, Toulouse, France; [27]PAASP France, Toulouse, France; [28]Maj Institute of Pharmacology, Polish Academy of Sciences, Krakow, Poland; [29]Institute of Pharmacology, Toxicology and Pharmacy, Ludwig-Maximilians-University, Munich, Germany; [30]Personalized Medicine Laboratory (LAMPER), Research Center

*For correspondence: anton.bespalov@paasp.net

†These authors contributed equally to this work

Present address: ‡ERBC Group, Baugy, France

inTechnology and Design Assistance of Jalisco State, National Council of Science andTechnology (CIATEJ-CONACYT), Mexico, Mexico; [31]Scripps Research Translational Institute, La Jolla, United States; [32]Integrative Structural and Computational Biology, Scripps Research, La Jolla, United States; [33]Neuro-BAU, Department of Fundamental Neurosciences, Faculty of Biology and Medicine, University of Lausanne, Lausanne, Switzerland; [34]Institute of Medical Sciences, University of Aberdeen, Scotland, United Kingdom; [35]SYRCLE, Department for Health Evidence, Radboud University Medical Center, Nijmegen, Netherlands; [36]Department for Clinical Medicine, Aarhus University, Aarhus, Denmark; [37]Institute of Pharmacology, Medical Faculty, University of Belgrade, Belgrade, Serbia; [38]Institute of Science and Technology, Klosterneuburg, Austria; [39]Neuroscience Center and Laboratory Animal Center, Helsinki Institute of Life Science, University of Helsinki, Helsinki, Finland; [40]Pain Research, Department of Surgery and Cancer, Faculty of Medicine, Imperial College London, London, United Kingdom; [41]Avertim, Brussels, Belgium; [42]Centre for Clinical Brain Sciences, University of Edinburgh, Scotland, United Kingdom

**Abstract** While high risk of failure is an inherent part of developing innovative therapies, it can be reduced by adherence to evidence-based rigorous research practices. Supported through the European Union's Innovative Medicines Initiative, the EQIPD consortium has developed a novel preclinical research quality system that can be applied in both public and private sectors and is free for anyone to use. The EQIPD Quality System was designed to be suited to boost innovation by ensuring the generation of robust and reliable preclinical data while being lean, effective and not becoming a burden that could negatively impact the freedom to explore scientific questions. EQIPD defines research quality as the extent to which research data are fit for their intended use. Fitness, in this context, is defined by the stakeholders, who are the scientists directly involved in the research, but also their funders, sponsors, publishers, research tool manufacturers, and collaboration partners such as peers in a multi-site research project. The essence of the EQIPD Quality System is the set of 18 core requirements that can be addressed flexibly, according to user-specific needs and following a user-defined trajectory. The EQIPD Quality System proposes guidance on expectations for quality-related measures, defines criteria for adequate processes (i.e. performance standards) and provides examples of how such measures can be developed and implemented. However, it does not prescribe any pre-determined solutions. EQIPD has also developed tools (for optional use) to support users in implementing the system and assessment services for those research units that successfully implement the quality system and seek formal accreditation. Building upon the feedback from users and continuous improvement, a sustainable EQIPD Quality System will ultimately serve the entire community of scientists conducting non-regulated preclinical research, by helping them generate reliable data that are fit for their intended use.

## The challenge: discovery of novel therapies requires rigor in research practices

The success rate in the discovery of novel, safe and effective pharmacotherapies has been declining steadily over the last few decades (*Scannell et al., 2012*). There are several factors likely accounting for this unfortunate record (*DiMasi et al., 2016*; *Waring et al., 2015*; *Shih et al., 2018*). While some of these factors (e.g. deeper knowledge of disease biology or clinical trial methodology) will take years, if not decades, of continued research to be properly addressed, others can be readily controlled today (*Bespalov et al., 2016*; *Landis et al., 2012*). One area requiring immediate attention is research rigor, which is estimated to be lacking in 50–90% of preclinical studies (*Freedman et al., 2015*).

High risk of failure is an inherent part of developing innovative therapies (*DiMasi et al., 2016*). However, some risks can be greatly reduced and avoided by adherence to evidence-based rigorous research practices. Indeed, numerous analyses conducted to date have clearly identified measures that need to be taken to improve research rigor (*Begley and Ioannidis, 2015*; *Landis et al., 2012*; *Ritskes-Hoitinga and Wever, 2018*; *Vollert et al., 2020*; *Volsen and Masson, 2009*).

## The EQIPD consortium: enhancing research quality as the main objective

Improving research rigor has biomedical, societal, personal, economic and ethical benefits for academia and industry alike, since the development of novel therapies is often rooted in academic discoveries and requires a highly specialized effort of industry to translate these discoveries into clinically useful applications. Moreover, the simple dichotomy between purely academic research and large industry/big pharma efforts is currently being replaced by networks of biotechs, spin-offs, private and public funders, contract research organizations (CROs), academic institutions engaging in drug discovery projects and manufacturers of research tools. It is therefore important that strategies to increase the robustness and reliability of preclinical research, both in terms of conduct and reporting, involve all these different stakeholders.

To address this challenge in preclinical biomedical research in a collaborative manner, the Enhancing Quality in Preclinical Data (EQIPD; originally called European Quality in Preclinical Data) consortium was formed in 2017 with founding members from 29 institutions across eight different countries (https://quality-preclinical-data.eu). The consortium works closely with a large group of associated collaborators, advisors and stakeholders representing research institutions, publishers, funders, learned societies and professional societies, from more than 100 organizations in Europe and the US.

Supported through the European Union's Innovative Medicines Initiative (IMI), the EQIPD consortium, among other deliverables, aimed to develop a novel preclinical research quality system that can be applied in both the public and private sectors. Such a quality system should be suited to boost innovation by ensuring the generation of robust and reliable preclinical data while being lean, effective and not becoming a burden that could negatively impact the freedom to explore scientific questions.

EQIPD defines research quality as the extent to which research data are fit for intended use (for related definitions and explanations, see *Juran and Godfrey, 1999*; *Gilis, 2020*). Fitness, in this context, is defined by the stakeholders, who can be scientists themselves, but also patients, funders, sponsors, publishers, and collaboration partners (e.g. peers in a multi-site research project).

The EQIPD consortium has developed a quality system that is free for anyone to use. Further, EQIPD is preparing training, support and assessment services for those research units that successfully implement the quality system and would like to seek formal accreditation.

## A *new* quality system to boost innovation

Quality systems usually appear as a response to an existing need (*Table 1*). For example, the development of the Good Laboratory Practice (GLP) standards, introduced first by the Food and Drug Administration (FDA) in the late 1970s, was triggered by poor research practices that compromised human health, such as mis-identification of control and experimental animals, omitted, non-reported or suppressed scientific findings, data inventions, replacements of animals lost to follow-up, and misdosing of animals (*Bongiovanni et al., 2020*; *Marshall, 1983*). In the Organisation for Economic Cooperation and Development (OECD) Principles (https://www.oecd.org/chemicalsafety/testing/overview-of-good-laboratory-practice.htm), GLP is defined as 'a quality system concerned with *the organisational process and the conditions* under which non-clinical health and environmental safety studies are planned, performed, monitored, recorded, archived and reported'.

GLP is a standard approach to quality in the regulated areas of preclinical drug development (which largely relate to non-clinical safety and toxicology studies rather than efficacy; see Appendix 1 Glossary for a definition of regulated research), where trained personnel perform mainly routine analyses, following defined Standard Operating Procedures (SOPs), and deliver data ultimately supporting patient safety.

**Table 1.** Comparison of quality systems.

| Quality system | ISO 9001 | GLP (FDA, OECD) | EQIPD |
|---|---|---|---|
| Year Launched | 1987, 2015 | 1976, 1981 | 2020 |
| Application area | A general QMS that can be applied to all aspects of organizations (not focused on biomedical research) | Non-clinical health and environmental safety studies upon which hazard assessments are based | Non-regulated preclinical (non-clinical) biomedical research |
| Initial stimulus to be developed | Procuring organizations needed a basis of contractual arrangements with their suppliers (i.e., basic requirements for a supplier to assure product quality) | Regulators such as FDA aimed to avoid poorly managed or fraudulent non-clinical studies on safety of new drugs | Biomedical research community (industry and academia) recognized the negative impact of lacking research rigor on the development of novel therapeutics, and the need for a comprehensive practical solution to help enhance preclinical data reliability |
| Customers | Typically outside of the organization (anyone who requires a product or service) | Typically outside of the organization (patients, regulators, sponsors, etc.) | In most cases, both inside (scientists themselves) and outside (patients, funders, collaboration partners, publishers, etc.) of the organization |
| Objectives | To certify that a product (which can be preclinical data) or a service is provided with consistent, good-quality characteristics, which satisfy the stated or implied needs of customers | To ensure the quality, integrity and reliability of data on the properties and/or safety of test items concerning human health and/or the environment | To facilitate generating robust and reliable preclinical data and thereby boost innovation |
| Main focus | Standardization of processes The organizational overall performance is continuously improved (process approach) to enhance customer satisfaction and development initiatives are done on a sound basis for sustainability | The organizational process and the conditions under which non-clinical health and environmental safety studies are planned, performed, monitored, recorded, archived and reported | The outcome of research activities that is robust, reliable, traceable, properly recorded, reconstructible, securely stored and trustworthy (generated under appropriately unbiased conditions) |
| Dedicated quality professionals | Not required (advisable for larger organizations) | Required | Not required (advisable for larger organizations) |
| Formal training on implementation and use | Not required | Required | Advisable, but not required |
| Assessments | External (ISO auditors) and internal (internal auditors) | External (health authorities/governmental inspectors) and internal (QA auditors) | Self-assessment (by Process Owner), external (by EQIPD)[*] |

*Additional internal assessments may be conducted by qualified colleagues (e.g. dedicated quality professionals) outside the research unit but within the same organization (advisable for larger organizations).

There have been attempts to develop a quality system based on GLP, that is, taking GLP as the basis and eliminating elements that are seen as excessive for the purposes of non-regulated drug discovery. However, GLP does not provide explicit guidance regarding those aspects of study design, conduct, analysis and reporting that are important to minimize the risk of bias and make research robust. In other words, even if it were made less demanding, conventional GLP cannot address some of today's key challenges in non-regulated preclinical research.

In contrast, the EQIPD Quality System is a novel system specifically aimed at supporting innovation in preclinical biomedical research. While the direct consequence of installing a quality system will be the generation of research data that are of higher rigor, the ultimate goal is to improve the efficiency of developing novel effective and safe therapies.

## Development of a new quality system by EQIPD

EQIPD was started in October 2017 and during the first phase (until June 2018), three work packages of the EQIPD consortium have delivered:

- A systematic review of guidelines for internal validity in the design, conduct and analysis of research involving laboratory animals (*Vollert et al., 2020*);
- An inventory of current practices and expectations toward quality management in non-regulated preclinical research (based on interviews with 70 consortium members and stakeholders);

- A review and analysis of governance in existing quality management systems (AAALAC International; ASQ Best Quality Practices for Biomedical Research in Drug Development; BBSRC Joint Code of Practice; ISO 9001, ISO 17025, ISO 15189; Janssen discovery quality system; Novartis research quality system; OECD Principles of GLP; RQA – Quality Systems Workbook).

During the second phase (July 2018 - January 2019), a working group was assembled from the EQIPD consortium members (n = 20). Based on the collected information, the working group nominated 75 statements that could define a functional quality system in non-regulated research. After three Delphi feedback rounds and two consensus meetings, these statements were revised, resulting in a final list of 18 core requirements (*Table 2*; see below for details).

During the third phase (February 2019 – September 2019), a supporting framework was developed (see below) and pilot implementation of the quality system started at four independent research sites.

Based on the feedback from those pilot implementation sites and interactions with the stakeholder group, an updated version of the framework was released for beta-testing in November 2019. The final version of the quality system was released in September 2020.

## The EQIPD Quality System: key features

*Table 3* presents five principles on which the EQIPD Quality System is based. These principles delineate in a maximally concise and direct form that the EQIPD Quality System is meant to support scientists in triggering changes in research practices, and that it will help to identify objectives and direction of change but will not prescribe any specific solutions as long as the research processes are kept transparent and traceable.

The EQIPD Quality System will deal with highly diverse research environments and associated challenges. The five principles are, therefore, instrumental in finding answers to specific questions – for example, is this particular practice in line with the EQIPD expectations? or should this particular process be documented?

**Table 2.** EQIPD Core Requirements.

| Categories | # | Item |
|---|---|---|
| Research team | 1 | Process Owner for the EQIPD Quality System must be identified |
| | 2 | Communication process must be in place |
| Quality culture | 3 | The research unit must have defined quality objectives |
| | 4 | All activities must comply with relevant legislation and policies |
| | 5 | The research unit must have a procedure to act upon concerns of potential misconduct |
| Data integrity | 6 | Generation, handling and changes to data records must be documented |
| | 7 | Data storage must be secured at least for as long as required by legal, contractual or other obligations or business needs |
| | 8 | Reported research outcomes must be traceable to experimental data |
| | 9 | Reported data must disclose all repetitions of a study, an experiment, or a test regardless of the outcome |
| Research processes | 10 | Investigator must declare in advance whether a study is intended to inform a formal knowledge claim |
| | 11 | All personnel involved in research must have adequate training and competence to perform assigned tasks |
| | 12 | Protocols for experimental methods must be available |
| | 13 | Adequate handling and storage of samples and materials must be ensured |
| | 14 | Research equipment and tools must be suitable for intended use and ensure data integrity |
| Continuous improvement | 15 | Risk assessment must be performed to identify factors affecting the generation, processing and reporting of research data |
| | 16 | Critical incidents and errors during study conduct must be analyzed and appropriately managed |
| | 17 | An approach must be in place to monitor the performance of the EQIPD Quality System, and address identified issues |
| Sustainability | 18 | Resources for sustaining the EQIPD Quality System must be available |

**Table 3.** Key principles.

| Principle | Explanation | Examples (related to the use of randomization) |
|---|---|---|
| Engage with autonomy | Decisions about specific needs and solutions are made by researchers, and not by EQIPD. EQIPD has formulated core requirements for the QS implementation and, as a partner in this process, EQIPD asks critical questions and provides recommendations that are voluntary to follow and are provided only to help the researchers throughout the implementation and use. | EQIPD recommends applying randomization to all studies but it is for the researcher to decide whether randomization is applied to a particular study or a particular study design |
| Grow through reflection | What it means to have the right quality level in place is suggested by your environment (collaborators, funders, institution, etc.). EQIPD does not 'invent' needs or requirements of your funders or your collaborators. As a partner in this process, EQIPD QS only allows you to see these requirements better and suggests ways of implementing them (*Gilis, 2020*). | EQIPD identifies overlapping requirements from different stakeholders toward the use and reporting of randomization. |
| Focus on goal | Focus on the outcome (performance standards), not on the path, timelines or the tools to get there (*Guillén, 2010*). | EQIPD highlights the importance of 'randomness' (lack of pattern or predictability) in the correctly developed randomization sequence but leaves it up to the user to select a specific method or tool. |
| Be transparent | Key research processes must be transparent. This principle applies specifically to retention and accessibility of information related to key decisions related to study design, conduct, or analysis (e.g. decisions to include or exclude certain data points in the analysis). | If one decides not to apply randomization, the decision must be stated and must be justified, recorded and reported. |
| Leave a trace | Key research processes must be traceable. Complementary to the principle above, this principle refers to retention and accessibility of all information that is necessary for a complete reconstruction of a key research process (e.g. raw data related to reported data are findable, and reported data are reconstructable from raw data). | If one does apply randomization, the way you apply randomization must be traceable and reported. |

## Flexible: driven by the needs of an individual research unit

Research environments are highly diverse: the needs of researchers at a big pharma company are different from those at a biotech; the needs of CROs are different from those of academic labs, etc. Thus, improving data quality is a challenge that cannot be tackled using a one-size-fits-all solution and flexibility is a critical requirement for future success.

The EQIPD Quality System is flexible: researchers are not confronted with a long and definitive A-to-Z list of what should be done and in what sequence. Instead, implementation of the EQIPD Quality System is characterized by:

- user-specific content – that is the exact nature of the individual elements of the EQIPD Quality System are defined largely by the users and their environment;
- a variable trajectory – that is there are very limited expectations regarding the sequence of introducing the different elements of the EQIPD Quality System; and
- no deadlines or fixed timelines – that is each unit adopts the EQIPD Quality System at its own pace, depending on the existing needs and available resources.

EQIPD has developed tools (for optional use) that help scientists to identify and organize information to address their own customized needs (e.g. related to *my* research funding source, *my* national regulations for the use of animals, expectations of *my* collaboration partners, policies set by *my* institution, *my* own commitment to research rigor, etc.). Being unique to a research unit or a researcher, such needs can be very specific to local or personal circumstances (i.e. essential for *my* success, *my* funding, *my* career, for instance because of the requirements of *my* preferred funder), and as such may be addressed with a higher or lower priority. Based on these factors, each research unit or researcher can determine their sequence of actions (*Figure 1*). EQIPD tools offer examples and ready-to-use solutions as well as information to develop new user-specific solutions.

For example, EQIPD has reviewed research quality expectations of several major public funders and pharmaceutical companies. Summaries of these expectations as well as examples of how these expectations can be met are available for downloading from the EQIPD's online Toolbox (https://eqipd-toolbox.paasp.net).

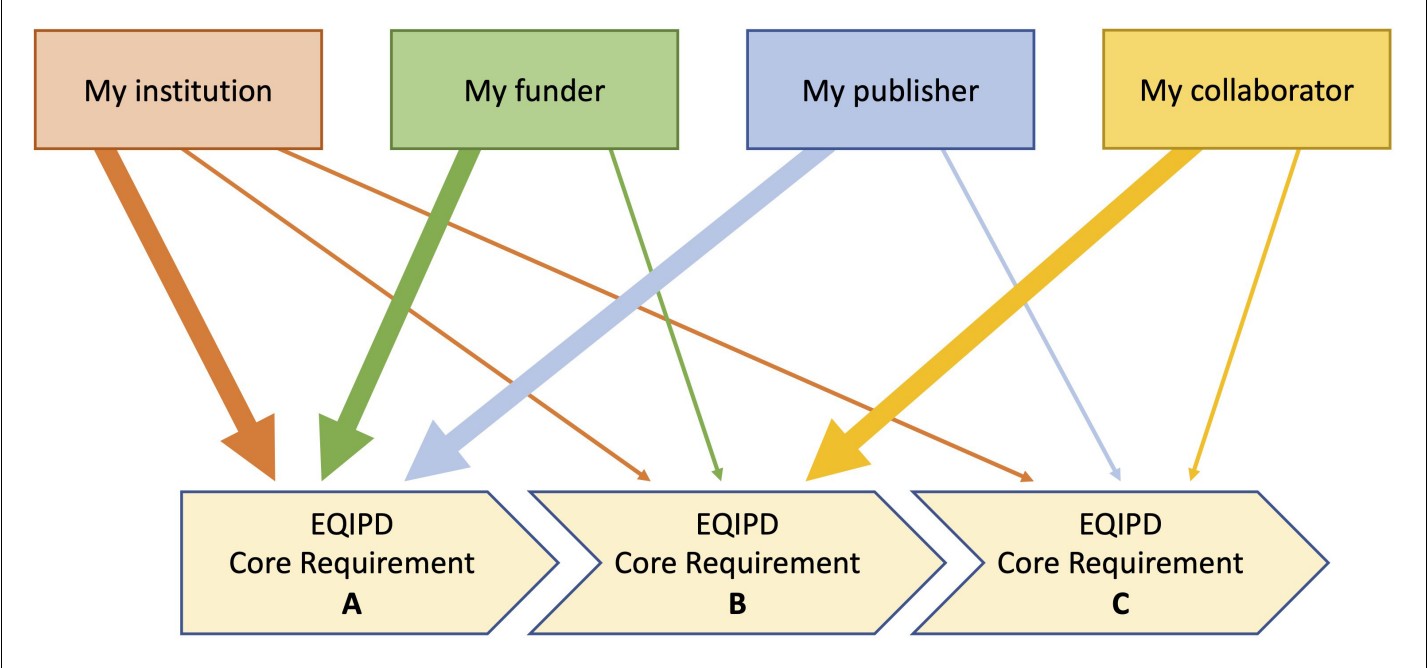

**Figure 1.** Flexible sequence of implementation of the EQIPD core requirements. Depending on the current needs, a research unit may prioritize the implementation of one or another core requirement. For example, tasks related to core requirement 'B' are highly relevant for the research unit's parent institution, the funding organization and a scientific journal where the research team plans to publish the results of their work. In contrast, core requirement 'C' is of lower importance and can, therefore, be addressed at a later timepoint.

### Team effort: understanding and endorsing research quality objectives

The focus on the specific needs of an individual research unit is ensured by the *Process Owner*, a person within the organization who has access to the necessary resources, and the competence and the authority to implement all steps needed to establish the EQIPD Quality System. Typically, the Process Owner should be someone who directs the work of the research unit (e.g. group leader, principal investigator, CEO or department head) and is knowledgeable about the importance of quality in research. EQIPD expects the Process Owner to be identified at the very first step of implementing the EQIPD Quality System (*Table 2*; core requirement #1).

In the second step, the Process Owner defines the *scope* - that is, the research unit (lab, territory, organization or part thereof) where the EQIPD Quality System will be applied - and identifies colleagues who will be actively involved in working on the implementation, as well as those who will be informed and may need to be trained about the new process (core requirement #2; *Table 2*). To that end, the Process Owner sets up a *communication plan* to support the team's buy-in and to facilitate a two-way information flow, in order to also capture feedback related to performance of the existing and newly introduced practices.

EQIPD also expects research units to define *quality objectives* (core requirement #3; *Table 2*). Although it may sound formal, this core requirement is indispensable and should be articulated at a level understandable and meaningful to everyone in the research unit.

Why are quality objectives needed? Once the Process Owner has decided to accept the role and responsibilities and has defined the research unit where the EQIPD Quality System will be implemented, it is worth getting prepared to answer questions that will likely come from colleagues inside and outside of the research unit: why are we doing this if, at least today, no such quality system is required by funders or collaboration partners and if, at least on first sight, we can successfully meet the goals without changing anything?

The answer to these questions helps to justify the efforts and time to be invested in the implementation and maintenance of the quality system. It also provides an argument by balancing the potentially negative impact on traditional metrics of scientific success (e.g. fewer positive results

generated, more time needed to complete projects) against the value of higher quality research (greater confidence in the results and scientific interpretations when results are shared with peers or published, improved rigor in decision making, more robust publications that stand the test of time, etc.).

In EQIPD terms, the answer should be documented as a *mission statement*, that is, a concise summary of why quality matters for that specific research unit. EQIPD provides examples of how scientists working in different roles and at various types of organizations may answer the question 'why quality matters' (https://eqipd-toolbox.paasp.net/wiki/Why_quality_matters).

It is important that the mission statement is understood, willingly accepted and followed by all members of the research unit.

If a Process Owner, alone or together with the research team members, cannot generate a clear and convincing answer to this question, no further steps should be taken and the implementation of the quality system is best postponed until a good answer is found and the research team is willing to embrace a quality mindset.

## EQIPD Quality System as part of the overall organizational quality culture

The Process Owner may also be asked and should be prepared to explain that the EQIPD Quality System does not replace and does not intend to re-interpret any of the existing rules, policies and other quality systems (which focus on specific areas) that apply to the research unit's environment.

EQIPD mandates that 'all activities must comply with relevant legislation and policies' (core requirement #4; *Table 2*) and that a 'research unit must have a procedure to act upon concerns of potential misconduct' (core requirement #5; *Table 2*). For the vast majority of organizations, no additional effort will be required to meet these expectations. If so, why are they included in the list of core requirements?

First, EQIPD does not want to be associated with organizations that engage in or tolerate unacceptable ethical practices or legal violations.

Second, the EQIPD Quality System is focused on quality, not legislation. Legislation may differ from country to country and for different research activities; hence, it is not possible to specify these individually in the EQIPD Quality System. Furthermore, EQIPD cannot oversee the way an organization deals with the legal requirements of, for example, handling hazardous substances, but emphasizes the need for compliance with such regulations as a basis on which all other quality measures rest.

A particularly relevant example concerns the care and use of laboratory animals that play a pivotal role in the research process. Society has granted the biomedical research community with the privilege to use laboratory animals in research under very specific conditions, all aiming to prevent inappropriate use of these ethically highly sensitive resources. Clearly, it is not acceptable to waste animals due to poor study design, conduct or analysis.

Ethical concerns on the use of animals in research have promoted the creation of a legal framework in almost every country (e.g. Animal Welfare Act in the US; Directive 2010/63 in the EU). Scientific evidence demonstrates that many aspects of animal care and use that are beyond the legal requirements have a direct impact on research results (*Guillén and Steckler, 2020*). The EQIPD team has developed a concise checklist that allows scientists to review if their animal care and use processes meet at least a minimum standard that supports the implementation and maintenance of the EQIPD Quality System. This review could optionally serve as the basis for further, more specific accreditation of the animal care and use program (i.e. AAALAC International accreditation) to ensure the implementation of high standards of animal care and use that would further contribute to increasing the quality of research (Appendix 2 Animal care and use checklist).

## EQIPD-defined principles, user-defined content

Implementation of the EQIPD Quality System does not require researchers to stop or reduce ongoing experimental work. It is designed so that it takes only minimal effort to sign up and begin the journey toward a quality system that should help researchers gradually improve certain quality aspects of their work.

The EQIPD Quality System gives guidance on expectations for quality-related measures, defines criteria for adequate processes (i.e. performance standards; see Appendix 1 Glossary for definition) and provides examples of how such measures can be developed and implemented. However, it does not prescribe any pre-determined solutions. Rather, users define their own specific solutions tailored to their individual settings.

For example, integrity of research data is one of the central concepts that the EQIPD Quality System aims to support. Four core requirements define the desired outcomes for raw data generation and handling (core requirement #6; *Table 2*), data storage (core requirement #7; *Table 2*), data traceability (core requirement #8; *Table 2*), and transparency of reported data (core requirement #9; *Table 2*). Thus, the 'what' is clearly described. However, there are various ways to fulfil these requirements. For instance, secure data storage could be achieved by using conventional paper-based laboratory notebooks, electronic laboratory notebooks, custom-built electronic solutions or paper-based controlled-access archives. Thus, there is flexibility in how integrity of research data could be achieved, and it is for the users of the system to decide on the best solution for their specific situation.

## Focused on the generation of fit-for-purpose research data

In general, EQIPD recommends that scientists apply protection against risks of bias for every study and unambiguously disclose the protective measures used. Each study has a particular purpose and the rigor applied to the study should be defined, documented in advance and be commensurate with the purpose of the study.

There are modes of research that can tolerate a certain level of uncertainty and do not lead to a formal *knowledge claim* (see Appendix 1 Glossary for definition). Such work is an essential part of the research process and may be used to generate hypotheses or to provide evidence to give the investigator greater confidence that an emerging hypothesis is valid, to develop new methods or to 'screen' compounds for potential effects prior to more formal testing.

There are also modes of research where researchers cannot accept inadequate control of the risks that can bias the research results (*Dirnagl, 2016*; *Hooijmans et al., 2014*). For research that is conducted with the prior intention of informing a knowledge claim, EQIPD requires that maximal possible rigor is applied (and exceptions explained and documented in the study plan; see *Table 4*). Such research will usually (but not always) involve some form of null hypothesis statistical testing or formal Bayesian analysis. Here, hypotheses are articulated in advance of data collection, with pre-specified criteria defining the primary outcome measure and the statistical test to be used.

Examples of research requiring maximal possible rigor include:

**Table 4.** Expectations toward rigor in study design.

|  | **All research** | **Research informing a formal knowledge claim (i.e. research requiring maximal rigor)** |
| --- | --- | --- |
| Study plan | Should be defined and documented before starting the experiments | Must be defined and documented before starting the experiments |
| Study hypothesis | Advised to define | Must be pre-specified |
| Blinding | Advised to implement | Should be implemented, exceptions must be justified and documented |
| Randomization | Advised to implement | Should be implemented, exceptions must be justified and documented |
| Sample size calculation | Advised to define and document before starting the experiments | Must be defined and documented before starting the experiments (e.g. included in the study plan) |
| Data analysis | Advised to define and document before starting the experiments | Must be defined and documented before starting the experiments (e.g. as a formal statistical analysis plan and/or included in the study plan) |
| Inclusion and exclusion criteria | Advised to define and document before starting the experiments | Must be defined and documented before starting the experiments (e.g. included in the study plan) |
| Deviations from study plan | Advised to document | Must be documented |
| Preregistration | - | Should be implemented |

- Experimental studies to scrutinize preclinical findings through replication of results (*Kimmelman et al., 2014*);
- Research aimed at generating evidence that enables decisions which will invoke substantial future investment (e.g. a decision to initiate a new drug development project or to initiate GLP safety assessment of a new drug candidate);
- Studies for which any outcome would be considered diagnostic evidence about a claim from prior research (*Nosek and Errington, 2020*);
- Labor-, resource-, and/or time-intensive studies that cannot be easily repeated.

EQIPD requires that investigators assert in advance whether a study will be conducted to inform a formal knowledge claim (core requirement #10; *Table 2*), and that they explicitly state this in the study (experimental) plans prepared before studies and experiments are conducted.

Further, it is required for all types of research that everyone in the research unit is adequately trained and competent (core requirement #11; *Table 2*), has access to protocols for experimental methods (core requirement #12; *Table 2*), follows adequate procedures for the handling and storage of samples and materials (core requirement #13; *Table 2*), and uses research equipment and tools that are suitable for the intended use (core requirement #14; *Table 2*).

## A system, not just a collection of guidelines and recommendations

Development and implementation of flexible and fit-for-purpose solutions are usually enabled by introducing a continuous improvement process (*Deming, 1986*). Within the EQIPD environment, the improvement cycle is rooted in the following workflow:

- Understand the rationale for introducing something new or modifying the current work routine (Why - the Need);
- Understand what is needed to achieve it (What - the Challenge);
- Propose a solution for achieving it (How - fit-for-purpose Solution);
- Evaluate the success of the implementation (Assessment).

As an example, a research organization is seeking a collaboration with a biopharmaceutical company (Why). The company informs the research organization about its expectations regarding the raw data record generation, handling, and storage. The research organization recognizes challenges associated with the storage of raw data as defined by the company (What). The EQIPD Toolbox provides information on what is the raw data and what are the best practices in recording and handling the raw data (How). In many cases, the new workflow is applied and has the desired effect. In some cases, there may be deficiencies identified that require remediation such as changes in the protocols, additional communication, educational and training efforts. Evaluation of the success in implementation of new processes concludes the cycle (Assessment).

In addition, the successful use of a new method or procedure often requires training, adequate and timely communication, feedback on incidents and errors, etc. To fully establish the EQIPD Quality System, several corrective or feedback mechanisms have to be included. These mechanisms identify factors affecting the generation, processing and reporting of research data *before* a study is done (core requirement #15; *Table 2*; see also *Box 1*), to analyze and manage the incidents and errors that may occur *during* the study (core requirement #16; *Table 2*), and to monitor the performance of the EQIPD Quality System (core requirement #17; *Table 2*; see also *Box 2*).

## Defining the user of the EQIPD Quality System

The ultimate mission of the EQIPD Quality System is to serve the entire community of scientists conducting non-regulated preclinical biomedical research. To achieve this goal, EQIPD's dissemination strategy *initially* focusses on early adopters, that is, research groups and scientists who:

1. See the value of higher standards of rigor in research to achieve more robust and reliable results, are willing to learn about and adopt a quality mindset and are prepared to invest effort to set up the EQIPD Quality System;
2. Consider their standards of rigor are already good, but strive to improve them further, and would like to establish the EQIPD Quality System as an independent *seal of quality*;

## Box 1. Managing risks to data quality .

Even under the best circumstances, not all recommended practices and protection measures can be applied to a working environment or research study, leaving a potential risk of failure. The EQIPD Quality System recognizes the following main areas where risk assessment should be conducted with risks made transparent and, if appropriate, documented:

1. Alterations from strongly recommended practices (i.e. situations in which the language of the EQIPD guidance includes 'should' and the research unit justifies why it does not or cannot apply). These assessments are done at regular intervals by the Process Owner;

2. Key and support processes that are inherently associated with risks endangering the validity of the results (e.g. risk of unblinding; emergency access to blinding codes). These assessments are done by scientists responsible for a study plan;

3. Changes in the environment *inside* of the research unit (changes in personnel; facility changes, etc.). These assessments are done or initiated *ad hoc* by the Process Owner.

4. Changes in the environment *outside* of the research unit (changes in personnel; facility changes, etc.). These assessments are done or initiated *ad hoc* by the Process Owner.

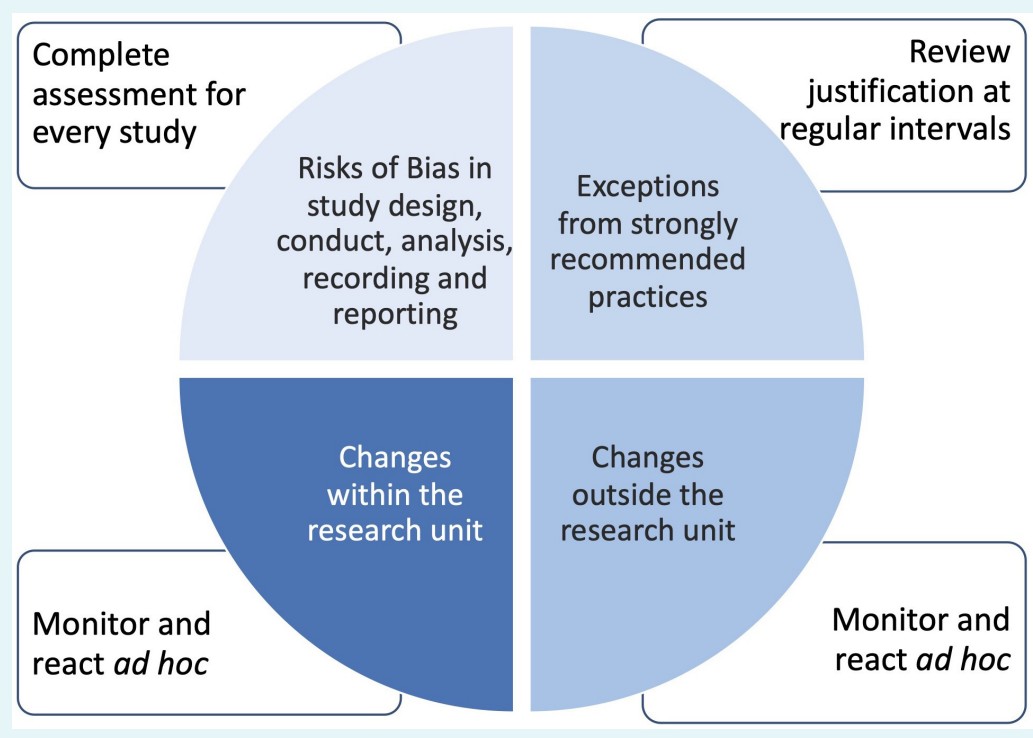

**Box 1—figure 1.** Areas of risk assessment.

3. Can use the EQIPD Quality System to strengthen a grant application, to support decision-making in drug discovery and/or to promote their services (e.g. CROs or academic labs active in the contract research domain) and bolster their reputation;

4. Are motivated by their funders, publishers and collaboration partners to secure high-rigor research standards (e.g. as a condition for funding or collaboration).

Such early adopters are known to be of critical value in every field where a cultural change is under discussion. For instance, academic initiatives have successfully addressed research data management and sharing of best practices by introducing Data Champions that serve as local advocates

# Box 2. Self-assessment.

The primary objectives of the self-assessment are to confirm that the research unit has everything in place for proper performance of the fit-for-purpose EQIPD Quality System, and to set the basis for internal or external quality checks/accreditation mechanism.

The Process Owner is responsible for defining the scope and frequency of this self-assessment, which is expected to involve all members of the research unit to ensure that all quality goals in the research unit have been considered and achieved.

As part of the self-assessment, there are spot checks conducted on selected documents (core requirements ## 11, 12, 16, 17; *Table 2*) and laboratory activities (core requirements ## 6, 7, 8, 9, 10, 13, 14, 15; *Table 2*). The Process Owner completes a paperless assessment of several solutions being up-to-date (core requirements ## 1, 2, 4, 5; *Table 2*), reviews and, if necessary, updates documentation (core requirements ## 2, 3, 6, 7, 8; *Table 2*), and engages the team in the discussion and review of certain processes (core requirements ## 3, 5, 13, 16; *Table 2*). The self-assessment itself is a core requirement (#17; *Table 2*) and can be conducted using a template provided in the Toolbox.

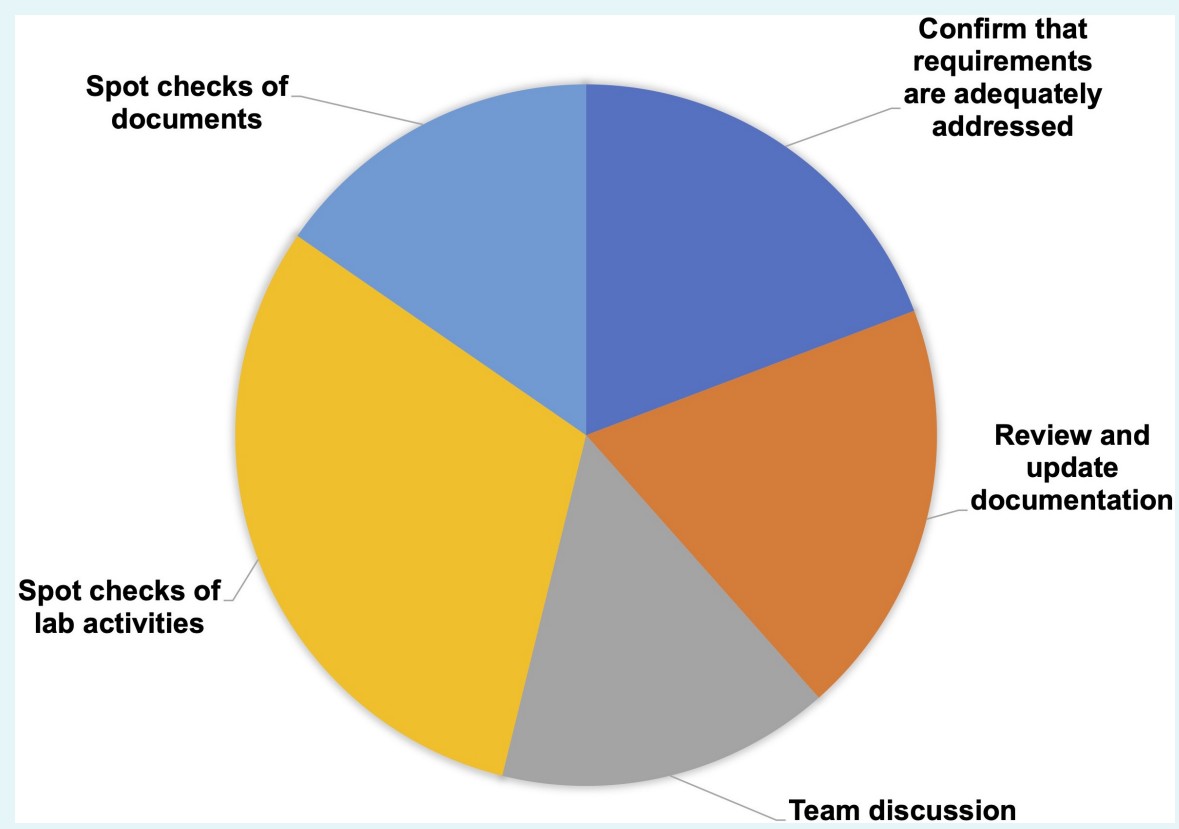

**Box 2—figure 1.** Types of self-assessment activities.

for good data practices (e.g. https://www.data.cam.ac.uk/intro-data-champions). Peer-to-peer learning eventually supports the dissemination of good practices beyond the early adopters.

The early adopters of the EQIPD Quality System, through their feedback to the EQIPD consortium, will help optimize the balance between the benefits of implementing such a system and any potential adverse consequences (e.g. resources allocated, reduction in conventional indices of scientific productivity). A positive balance will support further dissemination of the EQIPD Quality System

and help broader research communities take advantage of the work done by the EQIPD team and the early adopters.

It is a general understanding that not all research units are equally prepared or willing to implement a Quality System, an effort that requires investing time and resources. Tools developed and shared by EQIPD can also be used for other purposes – for example, as a source of information about specific aspects of good research practice, as a guidance for specific types of projects (e.g. industry-academia collaboration), or to enable a specific collaboration project by providing a purpose-fit certification of the current practices being in line with the EQIPD expectations (*Table 5*).

Since the scientists themselves will be the main users of the EQIPD framework, their leading and proactive role in improving the quality of their own scientific data will define the ways the framework can be used to prepare more and more research units to accept a Quality System as a means for long-term maintenance or research rigor standards.

## Implementation of the EQIPD Quality System

Even a lean and user-friendly quality system requires effort and resources to be implemented and maintained. This consideration makes it important to emphasize that a decision to start implementing the EQIPD Quality System should be well justified and regularly checked by the Process Owner and discussed with the research team.

### Size of the research unit

The EQIPD Quality System can be implemented at any level (university, research institute, company, or a laboratory). While this is the desired case, EQIPD encourages the transition toward better quality practices at the level of individual labs, departments or research groups, no matter how small they are, provided that there is a researcher capable, authorized and willing to take on the role of Process Owner.

The EQIPD Quality System is not intended to be used at the level of individual projects. Otherwise, it may create confusion and increase the risk of errors as the same people within a research unit may follow separate research quality practices depending on the project that they are working on.

**Table 5.** Levels of use of the EQIPD framework.

| Levels of use: | Information only (incl. training) | Purpose-fit certification | Quality System |
|---|---|---|---|
| EQIPD guidance: | Recommendations on best practices, examples, templates | Basic set of core requirements | Full set of core requirements |
| Main users: | Research units, funding organizations | Research units | Research units |
| Expected use: | As necessary, follow specific recommendations or use provided tools to improve work processes (e.g. increase transparency or make raw data findable or improve reporting)<br>As appropriate, use information provided by EQIPD in training programs; communicate to collaborators, grantees, etc. | Confirm that current quality practices are in line with the basic set of EQIPD core requirements (related to data integrity and rigor in study design, conduct, analysis, and reporting) | Align current research quality practices with the EQIPD expectations (implement full set of core requirements including those that define quality system – i.e. availability of resources, process owner, quality objectives, and continuous improvement mechanisms) |
| Dedicated efforts by the research unit (e.g., regular and sustained efforts, dedicated personnel): | None | Limited | Yes<br>(proportional to quality objectives) |
| Context of use: | Research unit is informed about expectations by current or future collaborators, funders, sponsors, publishers, etc. | Flexible solution driven by the time- and resource-critical needs of specific collaboration(s) | Stable solution for long-term maintenance of research rigor standards |
| Assessment by the EQIPD team: | No | Yes | Yes |

## Implementation path

There are several ways in which the EQIPD core requirements can be introduced within a research unit in terms of timing and sequence (*Figure 2*). Whether supported by the (optional) EQIPD tools or not, any of the possible implementation scenarios are acceptable as long as the outcome is the same, that is, a quality system implementing all 18 core requirements.

The implementation path suggested by EQIPD envisions three phases (Appendix 3 Implementation path):

Phase 1– A short list of cornerstone actions that are the same for all research units to help users understand why things are done, as well as ensuring that efforts triggered by the EQIPD framework have immediate impact (e.g. best practices to support data integrity and traceability).

Phase 2 – Users develop solutions for challenges directly connected to their environment or needs communicated by their funders, publishers and collaboration partners. During this phase, users meet most of the EQIPD core requirements while developing a habit of working toward a quality system.

Phase 3 – Completion of the remaining core requirements enabling formal recognition of a functional quality system.

The implementation is concluded with an important sustainability checkpoint: the Process Owner is expected to estimate the required resources and make them available for maintaining the EQIPD Quality System (core requirement #18; *Table 2*).

## Supporting tools

EQIPD has developed several tools (*Figure 2*) that are freely available to support the implementation and maintenance of the Quality System:

1. The Toolbox is a structured collection of information that enables users to build or select solutions for customized research needs. This Toolbox is built using wiki principles. The Toolbox contains a growing body of information about existing guidelines, recommendations, examples, templates, links to other resources, literature references, or just guidance on how to address a specific topic and will be regularly updated.

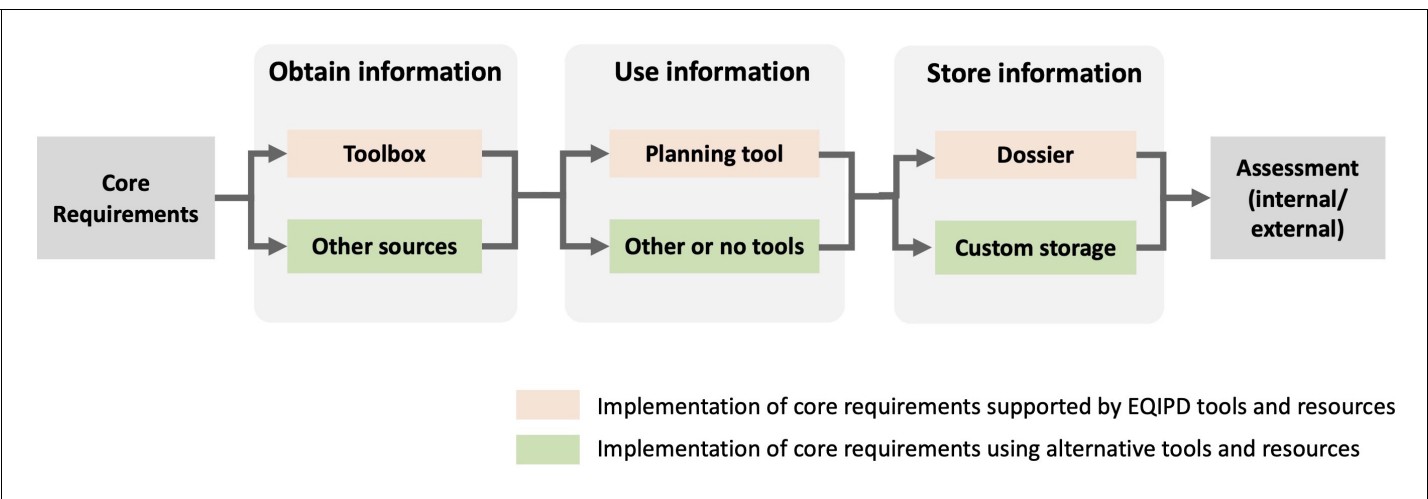

**Figure 2.** Implementation of the EQIPD Quality System (QS): From Core Requirements (CR) to assessment of a fully functional system. The 18 CRs are the expectations formulated by EQIPD that serve as the starting point for implementing the QS. At any step during the implementation, the use of EQIPD tools is voluntary and serves only the purpose of making the implementation and maintenance of the QS easier. As the first step, unless such information is available from other sources, the research unit may consult with the Toolbox to obtain relevant research quality-related information. Once the necessary information is obtained, the research unit applies this knowledge and monitors the progress. This can be done using the Planning Tool, using alternative project management resources or even without any such tools. The Dossier is a repository of documents and information that are specific to the user's research unit and that is organized according to a structure suggested by EQIPD (to keep all research quality-related information in one place and make it easily findable). However, the research unit may also opt to use its own way to store information. Finally, once the implementation is completed, the research unit may initiate an assessment to get feedback from experts outside of the research unit (either quality professionals within the same organization or a third party).

2. The Planning Tool is a user interface, designed to review the needs of researchers and is specific to their environment and focus of their research. Summarized expectations of funders, publishers, and collaboration partners can be entered in the Planning Tool either directly or using a special template called the Creator Tool.
3. The Dossier is a structured collection of customized documents and information related to research quality in a given research unit.

EQIPD does not intend to insist that researchers use these tools and rather sees their application as optional.

## The EQIPD Quality System: compliance mechanisms

The EQIPD system is a voluntary quality framework that enables research units to fulfill their own quality needs, for example, community guidelines or funder requirements.

Traditional quality systems require either internal (within the organization) or external auditors to check compliance with its system. This in turn requires that organizations employ dedicated and adequately trained quality professionals that understand the specific language in these quality regulations and ensure that the documentation formats correspond to the norm and nomenclature of the certifying organization.

The EQIPD Quality System is conceived as beginning with research scientists and extending to the research environment, and the compliance mechanisms are in line with this approach typically requiring no quality professionals.

### Self-assessment

The Process Owner is expected to use a self-assessment form provided by EQIPD to check whether Core Requirements and research unit-specific needs are appropriately addressed. The form guides the Process Owner through each core requirement, links out to the corresponding online Toolbox item, which describes background, expectations and provides further guidance documents.

The self-assessment serves two purposes. On the one hand, it allows the Process Owner to monitor performance of the quality system. On the other hand, it provides the base for an external assessment.

### External assessment

The external assessment of the research unit, performed as peer review, is a quality verification step that is recommended and important for the full implementation and the successful maintainance of QS. However, this step is not required and adopters of the Quality System may stop at the self-assessment stage.

External assessors review the self-assessment document and may request the research unit to provide additional documents. Assessors decide, based on the information provided, whether each core requirement is sufficiently addressed or whether additional verification is needed during the assessment interview.

The results of this preliminary assessment and further questions are shared with the research unit and are discussed in detail and clarified during the subsequent interview. A report is prepared by the assessors that details the results of the assessment, contains suggestions for improvement and ultimately confirms whether the research unit is compliant with all core requirements. Research units that successfully implemented the EQIPD Quality System receive a certificate of EQIPD compliance.

Several research units have completed the implementation of the EQIPD Quality System and have been evaluated by the EQIPD team.

External assessment is currently performed by scientists that developed the EQIPD Quality System. A training module for future assessors will be released to ensure the reliability and consistency of assessments conducted by different experts.

Moreover, anticipating a large demand for external assessments, the EQIPD team evaluates and compares the reliability of hybrid external assessment models combining onsite visits and remote interviews.

Importantly, EQIPD aims to make the assessment process as straightforward as possible. EQIPD's expectations are concisely summarized for each core requirement in a document that is regularly

updated and available via the Toolbox. Further, the EQIPD team advises to refer to the five key principles (*Table 3*) whenever a specific answer is not yet provided in the EQIPD guidance.

Last but not least, EQIPD's vision is that the Quality System serves the research units in the role of a partner, stimulating and guiding the continuous improvement in research rigor. With that in mind, EQIPD places a lot of weight on the competence and engagement of Process Owners conducting regular spot checks of key research processes and documentation.

## Enhancing Quality in Preclinical Data (EQIPD): the outlook

On September 30, 2020, the EQIPD Quality System was released for broad deployment and unrestricted use by the research community.

To enable the maintenance and further development of the EQIPD framework beyond the IMI project phase, the EQIPD team is implementing a governance model (*Figure 3*). The proposed model comprises three closely interacting levels:

- A strategic level represented by the EQIPD Guarantors, a group of the EQIPD project team members responsible for the overall guidance, administration of academic and educational programs, and the dissemination of the EQIPD vision. The EQIPD Guarantors will be supported by an Ethics and Advisory Board, a consultative body composed of current EQIPD consortium members, associate collaborators and advisors as well as key opinion leaders in the field of good research practice.

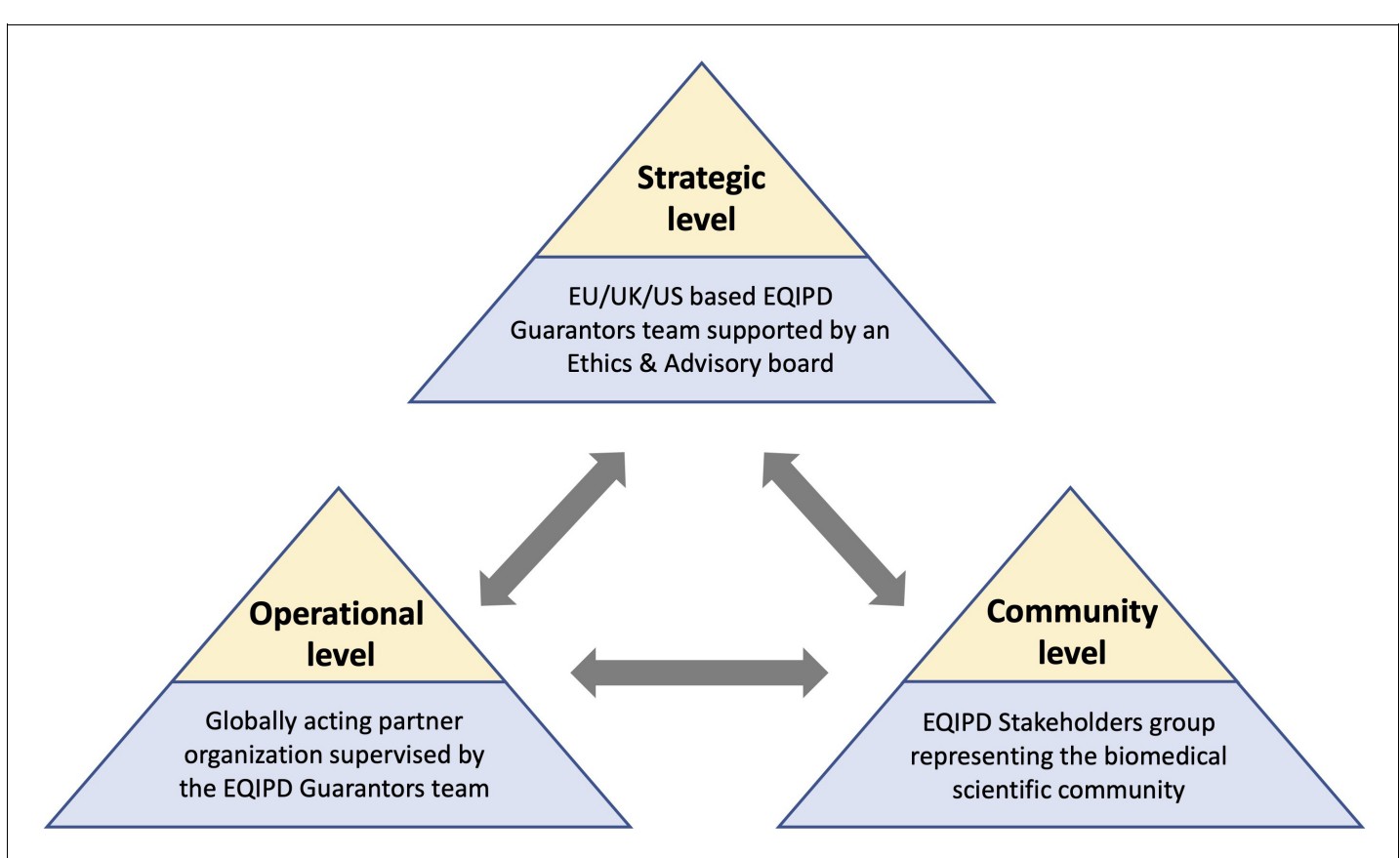

**Figure 3.** The proposed future governance model of EQIPD. The EQIPD Guarantors group and the EQIPD Ethics and Advisory Board are responsible for the overall guidance, administration of academic and educational programs, as well as dissemination of the EQIPD vision (*Strategic level*). An independent partner organization, commissioned by the EQIPD Guarantors, will provide the operational support and the day-to-day services for the EQIPD community (*Operational level*). The EQIPD Stakeholder group, composed of scientists, funders, quality professionals, manufacturers of research tools, and publishers, provides feedback on the practical aspects of the EQIPD Quality System and facilitates connections to a broader biomedical research community (*Community level*).

- An operational level represented by an independent globally acting partner organization, commissioned by the EQIPD Guarantors to provide the operational support and services required for day-to-day business management (including technical support and training for the research units during the implementation and maintenance of the EQIPD Quality System).
- A community level that is represented by the EQIPD Stakeholder group, a diverse group of scientists, funders, quality professionals, manufacturers of research tools, and publishers that provide feedback on practical aspects of the EQIPD Quality System and facilitates connections to a broader biomedical research community.

The next milestones for the EQIPD team are:

- Launch of an educational platform that will support both the use of the EQIPD Quality System and provide more general training in the field of good research practice;
- Analysis of geographical and cultural differences that may affect the acceptance of the EQIPD Quality System and that may require adaptations in the associated framework;
- Evaluation of the impact of implementation of the EQIPD Quality System on research quality, to inform further development of the EQIPD framework;
- The EQIPD Quality System was developed with the focus on the users and their needs. The EQIPD collaborators will maintain and expand this focus further.

The EQIPD team is actively engaged in discussions with funders (public and private) and publishers to develop instruments and mechanisms that will allow scientists to further benefit from the use of the EQIPD Quality System.

All scientists engaged in biomedical research are invited to join the growing community of the EQIPD Quality System users and supporters (http://www.eqipd.online).

# Acknowledgements

This project has received funding from the Innovative Medicines Initiative 2 Joint Undertaking under grant agreement No 777364. This Joint Undertaking receives support from the European Union's Horizon 2020 research and innovation programme and EFPIA. The authors are very grateful to Martin Heinrich (Abbvie, Ludwigshafen, Germany) for the exceptional IT support and programming the EQIPD Planning Tool and the Creator Tool and to Dr Shai Silberberg (NINDS, USA), Dr. Renza Roncarati (PAASP Italy) and Dr Judith Homberg (Radboud University, Nijmegen) for highly stimulating contributions to the discussions and comments on earlier versions of this manuscript. We also wish to express our thanks to Dr. Sara Stöber (concentris research management GmbH, Fürstenfeldbruck, Germany) for excellent and continuous support of this project. Creation of the EQIPD Stakeholder group was supported by Noldus Information Technology bv (Wageningen, the Netherlands).

# Additional information

### Competing interests

Anton Bespalov: AB is an employee and/or shareholder at PAASP GmbH, PAASP US LLC, Exciva GmbH, Synventa LLC, Ritec Pharma. Anja Gilis: AG are employees of Janssen / Johnson & Johnson and shareholders at Johnson & Johnson. Björn Gerlach, Christoph H Emmerich: BG and CE are employees and shareholders at PAASP GmbH. Javier Guillén: JG is an employee of AAALAC International that is an EQIPD Associated Collaborator. Vincent Castagné, Christelle Froger-Colléaux: VC and CFC are employees of Porsolt. Isabel A Lefevre, Fiona Ducrey: IAL and FD are employees of Sanofi. Lee Monk: LM is an employee and shareholder of UCB. Sandrine Bongiovanni: SB is an employee of Novartis Pharma. Bruce Altevogt: BA is an employee and shareholder of Pfizer. The views and opinions expressed in this article are those of the individual author and should not be attributed to Pfizer, its directors, officers, employees, affiliates, or any organization with which the author is employed or affiliated. Chantelle Ferland-Beckham: AB, BA, NdB, UD, CFB, PK, MK, MM, PM, PP, GR, JS, and TS are members of the Preclinical Data Forum (co-chairs - AB and TS), a network financially and organizationally supported by ECNP and Cohen Veterans Bioscience. Patricia Kabitzke: PK is an employee and shareholder at PAASP US LLC. Claudia Kurreck: UD and CK receive funding from Volkswagen Foundation. Paul Moser: PM is owner of Cerbascience Consulting. Heidrun Potschka: HP has received during the last three years consulting and speaking fees and/or funding

for collaborative projects from Bayer, Roche, Zogenix, and Eisai. Kathleen Wuyts: KW is a consultant of Avertim, Brussels, Belgium, support for this contribution was funded by Janssen Pharmaceutica NV. Malcolm R MacLeod: MM, UD and TS are members of the Advisory Board at PAASP. MM, UD and TS are members of the ARRIVE guidelines working group. Ulrich Dirnagl: UD and CK receive funding from Volkswagen Foundation. MM, UD and TS are members of the Advisory Board at PAASP. MM, UD and TS are members of the ARRIVE guidelines working group. Thomas Steckler: MM, UD and TS are members of the Advisory Board at PAASP. MM, UD and TS are members of the ARRIVE guidelines working group. TS is an AAALAC ad-hoc specialist. TS and AG are employees of Janssen / Johnson & Johnson and shareholders at Johnson & Johnson. The other authors declare that no competing interests exist.

## Funding

| Funder | Grant reference number | Author |
|---|---|---|
| Innovative Medicines Initiative | 777364 | Malcolm R MacLeod |

The funders had no role in study design, data collection and interpretation, or the decision to submit the work for publication.

## Author contributions

Anton Bespalov, Conceptualization, Data curation, Supervision, Funding acquisition, Validation, Investigation, Visualization, Methodology, Writing - original draft, Project administration, Writing - review and editing; René Bernard, Conceptualization, Data curation, Formal analysis, Supervision, Investigation, Methodology, Project administration, Writing - review and editing; Anja Gilis, Conceptualization, Resources, Data curation, Formal analysis, Supervision, Validation, Investigation, Visualization, Methodology, Writing - original draft, Project administration, Writing - review and editing; Björn Gerlach, Conceptualization, Resources, Data curation, Software, Formal analysis, Supervision, Validation, Investigation, Visualization, Methodology, Writing - original draft, Project administration; Javier Guillén, Conceptualization, Data curation, Formal analysis, Validation, Investigation, Methodology, Writing - original draft, Writing - review and editing; Vincent Castagné, Conceptualization, Data curation, Funding acquisition, Validation, Investigation, Methodology, Writing - review and editing; Isabel A Lefevre, Conceptualization, Data curation, Formal analysis, Investigation, Methodology, Writing - review and editing; Fiona Ducrey, Conceptualization, Validation, Investigation, Visualization, Methodology, Writing - review and editing; Lee Monk, Conceptualization, Data curation, Validation, Investigation, Visualization, Methodology, Writing - review and editing; Sandrine Bongiovanni, Conceptualization, Validation, Investigation, Methodology, Writing - review and editing; Bruce Altevogt, Conceptualization, Funding acquisition, Investigation, Methodology, Writing - review and editing; María Arroyo-Araujo, Alexander Dityatev, Raafat Fares, Valerie Gailus-Durner, Piotr Popik, Ernesto Prado Montes de Oca, Leonardo Restivo, Janko Samardzic, Michael Schunn, Claudia Stöger, Validation, Investigation, Writing - review and editing; Lior Bikovski, Chantelle Ferland-Beckham, Data curation, Validation, Investigation, Writing - review and editing; Natasja de Bruin, Martine CJ Hofmann, Validation, Investigation, Project administration, Writing - review and editing; Esmeralda Castaños-Vélez, Conceptualization, Data curation, Formal analysis, Validation, Investigation, Methodology, Writing - review and editing; Christoph H Emmerich, Conceptualization, Data curation, Validation, Investigation, Visualization, Methodology, Writing - original draft, Writing - review and editing; Christelle Froger-Colléaux, Data curation, Formal analysis, Validation, Investigation, Methodology, Writing - review and editing; Sabine M Hölter, Validation, Writing - review and editing; Patricia Kabitzke, Conceptualization, Resources, Validation, Investigation, Project administration, Writing - review and editing; Martien JH Kas, Supervision, Funding acquisition, Validation, Investigation, Project administration, Writing - review and editing; Claudia Kurreck, Conceptualization, Data curation, Validation, Investigation, Methodology, Writing - original draft, Writing - review and editing; Paul Moser, Validation, Investigation, Methodology, Writing - review and editing; Malgorzata Pietraszek, Data curation, Formal analysis, Validation, Investigation, Visualization, Methodology, Writing - review and editing; Heidrun Potschka, Gernot Riedel, Supervision, Funding acquisition, Validation, Investigation, Writing - review and editing; Merel Ritskes-Hoitinga, Conceptualization, Supervision, Validation, Investigation, Methodology, Writing - review and editing; Vootele Voikar, Validation,

Methodology, Writing - review and editing; Jan Vollert, Conceptualization, Validation, Investigation, Project administration, Writing - review and editing; Kimberley E Wever, Conceptualization, Supervision, Funding acquisition, Validation, Investigation, Project administration, Writing - review and editing; Kathleen Wuyts, Project administration, Writing - review and editing; Malcolm R MacLeod, Ulrich Dirnagl, Conceptualization, Resources, Supervision, Funding acquisition, Validation, Investigation, Methodology, Project administration, Writing - review and editing; Thomas Steckler, Conceptualization, Resources, Data curation, Supervision, Funding acquisition, Validation, Investigation, Visualization, Methodology, Writing - original draft, Project administration, Writing - review and editing

## Author ORCIDs

Anton Bespalov ⓘ https://orcid.org/0000-0003-3730-1395
René Bernard ⓘ https://orcid.org/0000-0003-3265-2372
Björn Gerlach ⓘ https://orcid.org/0000-0002-4900-6302
Javier Guillén ⓘ https://orcid.org/0000-0001-8188-7587
Vincent Castagné ⓘ https://orcid.org/0000-0002-8728-2922
Isabel A Lefevre ⓘ https://orcid.org/0000-0003-0086-5947
Sandrine Bongiovanni ⓘ https://orcid.org/0000-0002-8655-6514
Bruce Altevogt ⓘ https://orcid.org/0000-0002-1359-5416
María Arroyo-Araujo ⓘ https://orcid.org/0000-0001-6612-0679
Lior Bikovski ⓘ https://orcid.org/0000-0003-2275-1609
Natasja de Bruin ⓘ https://orcid.org/0000-0002-2456-2783
Esmeralda Castaños-Vélez ⓘ http://orcid.org/0000-0002-2670-1118
Alexander Dityatev ⓘ https://orcid.org/0000-0002-0472-0553
Christoph H Emmerich ⓘ https://orcid.org/0000-0002-7594-3973
Raafat Fares ⓘ https://orcid.org/0000-0003-3963-381X
Christelle Froger-Colléaux ⓘ https://orcid.org/0000-0003-0623-8928
Sabine M Hölter ⓘ http://orcid.org/0000-0003-4878-5241
Martine CJ Hofmann ⓘ https://orcid.org/0000-0002-8280-2036
Patricia Kabitzke ⓘ https://orcid.org/0000-0003-3305-0795
Martien JH Kas ⓘ https://orcid.org/0000-0002-4471-8618
Piotr Popik ⓘ https://orcid.org/0000-0003-0722-1263
Ernesto Prado Montes de Oca ⓘ https://orcid.org/0000-0002-0617-4752
Merel Ritskes-Hoitinga ⓘ https://orcid.org/0000-0001-5315-284X
Janko Samardzic ⓘ https://orcid.org/0000-0002-8464-4924
Michael Schunn ⓘ https://orcid.org/0000-0003-4326-5300
Claudia Stöger ⓘ https://orcid.org/0000-0001-8361-0274
Vootele Voikar ⓘ https://orcid.org/0000-0003-4201-8666
Kimberley E Wever ⓘ https://orcid.org/0000-0003-3635-3660
Malcolm R MacLeod ⓘ https://orcid.org/0000-0001-9187-9839
Ulrich Dirnagl ⓘ https://orcid.org/0000-0003-0755-6119

## Decision letter and Author response

Decision letter https://doi.org/10.7554/eLife.63294.sa1
Author response https://doi.org/10.7554/eLife.63294.sa2

# Additional files

## Supplementary files

• Transparent reporting form

## Data availability

We have not generated any data.

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

## Appendix 1

## Glossary

| Term | Definition |
|------|-----------|
| Challenge | An unmet requirement that must be appropriately matched by a specific solution. |
| Continuous improvement | The process ensuring that new risks and challenges are identified and appropriately met by adapting the existing quality system. |
| Core requirements | Tasks defined by EQIPD for all users/institutions that must be fulfilled to reach the status of a functional quality system. |
| Dossier (EQIPD Dossier) | A structured and categorized collection of various quality-related items (such as protocols for experimental methods and training records) that are developed and stored by a research unit as solutions to challenges specific to their needs. |
| Framework (EQIPD Framework) | The concept, implementation strategy, software and tools developed by EQIPD that direct and support the users in building the fit-for-purpose EQIPD Quality System. |
| Key process | An action or series of actions that directly impact the experimental generation of research products, data and their quality. |
| Knowledge claim | A formal knowledge claim can be thought of as a statement that a research project or study has established new knowledge, or consolidated existing knowledge, with sufficient certainty that that knowledge can now be acted upon. The required level of certainty might depend on the nature (risk and potential benefits) of the possible action. For instance, the required level of confidence in the efficacy of a molecule will be different for a decision to proceed to a clinical trial compared to deciding to publish the results of a study or to initiate a lead identification campaign for a newly validated target. |
| Needs | Reasons to introduce and maintain high quality derived from a research unit's mission and research objectives that can be dictated by stakeholders (e.g., funders) or defined by EQIPD. They are identified by the research unit. |
| Must | Indicates actions that EQIPD considers as imperative and mandatory or as a requirement. |
| Performance standards | Performance standards define the desired outcome in detail and provide measurable criteria for assessing whether the outcome is achieved, but do not specify a method or technique for achieving the desired outcome |
| Planning tool (EQIPD Planning tool) | A software tool supporting the research unit in order to implement the EQIPD Quality System in a given institution. |
| Process owner | A person within the organization/research unit who has the necessary resources or access to them, the competence and the authority to implement and maintain the EQIPD Quality System. |
| Regulated research, or research subject to regulation | Research activities for which national (e.g., FDA) or international (e.g., EMA, OECD) governmental bodies and agencies have specific responsibility for regulating the research activity as well as setting expectations and inspections. Regulated research is typically subject to compliance with the formally defined good practices such as Good Laboratory Practice, Good Manufacturing Practice, Good Clinical Practice, Good Pharmacovigilance Practice, etc. |
| Robust | A data set is said to be robust if it is not sensitive to departures from the assumptions on which its validity is strictly predicted (e.g., that the study plans used to generate data are protected against risks of bias). |
| Should | A strong recommendation; however, EQIPD recognizes that individual circumstances might justify an alternative strategy. |
| Solution | An answer to an identified challenge. |
| Support process | An action or series of actions that provide the means needed to execute key processes in a quality-oriented manner. |
| Toolbox (EQIPD Toolbox) | A structured and categorized online collection of various quality-related information, such as guidelines, protocols, and tools. |

### Appendix 2

## Animal care and use checklist

This checklist is intended for research units conducting studies using laboratory animals and not having a full accreditation from AAALAC (or equivalent). Guided by this checklist, research units are expected to develop a description of the animal care and use program.

1. Ethical evaluation and authorization process of animal use. The participation of the following bodies must be described:
   - Institutional body(ies) involved
   - External body(ies) involved
   - Competent Authority
2. Animal procurement and identification. The following topics must be defined and documented:
   - Source of animals
   - Transportation
   - Acclimation and quarantine procedure/periods
   - Acceptance criteria
   - Identification method
3. Animal housing conditions. The following topics must be defined and documented:
   - Description of caging/enclosure (dimensions, open, IVC, flooring, etc.)
   - Animal numbers by type of caging/enclosure
   - Environmental enrichment
   - Justified cases for single housing of social species relating to experimental procedures (e.g., metabolic caging, post-surgery), veterinary intervention or social incompatibility
4. Animal environmental conditions. The following topics must be controlled and recorded:
   - Air changes/hour in room and IVC
   - Air temperature ranges by species
   - Possibilities for thermoregulation at cage level
   - Ranges of relative humidity by species
   - Light source, intensity and cycle
5. Food, watering and bedding. The following topics must be defined and documented:
   - Feed source, type and treatment (e.g., autoclave, irradiation, etc)
   - Feed storage conditions
   - Feed/water provision (e.g., ad libitum, restricted)
   - Water source and treatment (e.g., acidification, chlorination, autoclave)
   - Water provision method (e.g., bottles, automatic) and frequency of change (in case of bottles)
   - Type of bedding and treatment (e.g., autoclave, irradiation)
   - Frequency of bedding change, and relation with experimental procedures if any
6. Sanitation procedure. The following topics must be defined and documented:
   - Frequency of change of microenvironment items: cage, lid, water bottle, etc.
   - Frequency of sanitation of macroenvironment: room level
   - Agents used
   - Monitoring effectiveness of sanitation of caging/enclosures
7. Frequency and procedure of observation of animals. The following topics must be defined and documented:
   - Personnel involved (animal care and/or research teams)
   - Reporting method to veterinarian and/or investigators
   - Implementation of defined humane endpoints
   - Response in case of unexpected outcomes
8. Animal health and genetic monitoring. The following topics must be defined and documented:
   - Health monitoring program and reports for the research colony
   - Scientific nomenclature of the species, with special emphasis on genetically modified animals
   - Method of control of genetic drift, if applicable
9. Veterinary interventions during the study. The following topics must be recorded:
   - Veterinary interventions and drugs used (e.g., analgesics, antibiotics, etc)
10. Surgical procedures. The following topics must be defined and documented:

- - Drugs used for surgical procedures
  - Aseptic technique
  - Dedicated room and equipment
  - Recovery procedures and post-surgical monitoring
11. Animal euthanasia procedures. The following topics must be defined and documented:
    - Euthanasia methods: default method and others according to type of experiment and species
    - Euthanasia conditions: separation from other animals, dedicated area
    - Methods of confirmation of death

## Appendix 3

### The EQIPD Quality System implementation path (optional)

Phase 1: Same for everyone

The first task is to establish a basis for building a quality system. This basis includes absolute must-have (core) elements that:

- are the same for all labs
- are required to support the next steps
- can be completed within days

This set of requirements was chosen according to the principles that are uniform to all types of research units and is completed by the following four steps:

Step 1. Establish a Process Owner

- WHY: This process needs a leader who takes the overall responsibility for implementing the EQIPD Quality System and is willing to drive the process forward.
- WHAT: EQIPD defines the Process Owner as a person within the research unit who has the necessary resources or access to them, the authority, the competence and takes the responsibility to start all implementation steps needed to establish the EQIPD Quality System.
- HOW: Different research units may have differing criteria for identifying such leaders, ranging from a self-nomination for a small research unit to dedicated professionals in larger organizations.

Step 2. Define quality objectives

- WHY: The EQIPD Quality System will turn into a burden and resources will be wasted if the Process Owner and the research team do not understand the objectives of investing time and efforts into the implementation and maintenance of the system.
- WHAT: The Process Owner, alone or in discussions with the research team, develops a concise summary of why quality matters for that specific research unit. The Process Owner discusses with the team what specifically is at stake if the proper quality is not maintained.
- HOW: EQIPD provides a template with guiding questions to develop a quality objective summary (called 'Mission') as well as examples of statements on how a quality system or higher research rigor could help scientists achieve goals in different environments and at different positions.

Step 3. Set up a communication plan in the research unit where the EQIPD Quality System is to be applied

- WHY: The EQIPD Quality System can only be implemented in a specific research environment where roles and responsibilities of every team member are clearly defined and there is a clear plan for communication between team members.
- WHAT: Define a research unit (lab, territory, organization or part thereof) where the EQIPD Quality System will be applied in order to identify all colleagues who need to be informed about the new process, colleagues who need to be actively involved in working on the EQIPD Quality System implementation; set up a communication plan/schedule of working meetings; make sure that the communication plan supports a two-way information flow (i.e., also to capture feedback related to the performance of the existing and newly introduced practices).
- HOW: Depending on the organization, the initial information can be distributed via email, the intranet or during regular or extraordinary team meetings. The Process Owner may assemble a dedicated team or appoint a colleague to be responsible for implementing the EQIPD Quality System. The Process Owner is responsible for establishing a Communication Plan. EQIPD provides guidance for establishing the communication plan and templates for documenting all roles and responsibilities. All these templates (as well as all other guiding and supporting information) can be found in the Toolbox.

Step 4. Establish a documentation plan to make data and associated documents traceable and any changes transparent

- WHY: Traceable and transparent handling of all information related to study design, conduct, analysis and reporting is a pre-requisite for good research practice.

- WHAT: The documentation plan should ensure the ability to find the source of data (raw and analyzed) that is presented in a report or presentation and should contain a contemporaneous description of the experiment from which the data originated. A qualified reviewer should be able to link figures, graphs, conclusions, and other summary data to the raw data that was processed/analyzed; link the summary data to the corresponding experiment described in a lab notebook entry; and, likewise, link the lab notebook entry to the raw data (e.g., when generated by an automated instrument).
- HOW: Depending on the organization, resources and workflows, this could be an electronic or paper-based lab notebook, a more complex Laboratory Information Management System (LIMS), or some other form of archiving as long as the information handling procedures are transparent and properly documented.

## Phase 2 – Defined by user-specific needs

During this Phase, the primary focus is on challenges that are specific to the research unit where the EQIPD Quality System is being established.

The team led by the Process Owner identifies needs pertinent to the researchers' environment – e.g., related to funding sources, collaboration partners, reporting and publication strategies, institutional, national and applicable international laws and regulations, use of animal subjects, etc.

Some of these needs have already been reviewed by the EQIPD team, summaries are available from the Toolbox.

Others will need to be reviewed by the Process Owner and the team to identify and develop appropriate solutions.

There are no time limits set to complete this phase. The organization may also remain in this position if, for example, there are not enough financial or personnel resources to continue the process. The decision to move to the next phase is made by the Process Owner and the team based on the following criteria:

- All high-priority challenges have been addressed;
- There are motivation, resources and organizational support to complete the implementation of the EQIPD Quality System.

## Phase 3 – Complete the system

The EQIPD team has designed a set of 18 Core Requirements (*Table 2*) that must be fulfilled in order to reach the status of a functional quality system. Several of these Core Requirements are addressed during Phase one and Phase 2.

The goal of Phase three is to address those Core Requirements that have not been implemented so far but are essential to complete the development of the EQIPD Quality System.

However, for different research units, the remaining effort will be different because the Core Requirements addressed during Phase two are selected/chosen depending on the user-specific needs.

Therefore, it is anticipated that there will be research units for which Phase three requirements will only be a small step, as most Core Requirements have already been implemented during Phase 1 and 2, and those for which this final step may be a significant task demanding time and resources.

EQIPD also explicitly acknowledges the possibility for research units to not pursue the completion of the System. In fact, going through Phase 1 and 2 in many cases may already enable high quality research outputs and researchers may be satisfied with the progress achieved and make an indefinite pause.

*****

In summary, the core requirements necessary to build and declare a functional EQIPD Quality System are spread over three implementation phases. It is expected that, in most cases, these requirements will not appear as a burden because:

- during Phase 1, the number of requirements is kept to an absolute minimum and the goal is only to ensure that the research team has everything needed in place to start building a Quality System;

- during Phase 2, requirements are mostly introduced indirectly – via research processes that are important and fully understandable for the users (i.e., while addressing the various needs specific to their environment, from funding and publishing to environmental health and safety);
- during Phase 3, research teams deal with the remaining requirements that may be easier to handle because:
    - a major portion of the work has already been done, and
    - there is a habit established that makes continuous improvement a manageable routine.

After the Quality System is established, the maintenance focus is on continuous improvement that accompanies the use of the Quality System. For example, there may be new needs and challenges identified.

