## [Decision Letter]

**Acceptance summary:**

The study reports the development of a novel quality system created under the aegis of the EQIPD Consortium. The focus of this validated toolkit is to support basic biomedical research, both in industry and academia. Major revisions and new data have significantly enhanced the quality of the conclusions and their application.

**Decision letter after peer review:**

Thank you for submitting your article "Introduction to the EQIPD Quality System" for consideration by *eLife*. Your article has been reviewed by 3 peer reviewers, and the evaluation has been overseen by Mone Zaidi as the Senior and Reviewing Editor. The following individual involved in review of your submission has agreed to reveal their identity: Michael Curtis (Reviewer #3).

The reviewers have discussed the reviews with one another and the Reviewing Editor has drafted this decision to help you prepare a revised submission.

Summary:

You describe the work from the Enhancing Quality in Preclinical Data (EQIPD) consortium, whose main goal was to develop a preclinical research quality system that can be applied in both public and private sectors and is non-proprietary. This constitutes a set of 18 principles, or "core requirements", that define objectives for the planning of research as well as for operations and analysis. Overall, the enthusiasm for this manuscript was modest and while all reviewers agreed that the article should be published, there were major suggestions for improvement.

Essential revisions:

The reviewers noted that the paper was noble for taking on the topic of false or misleading research findings and their goal to improve research rigor and honesty. Moreover, their introduction of a host of processes and box-checking steps to incorporate into research would be appreciated by lab managers. However, the reviewers note that, written in its current form as a process document, there were few concrete examples provided to illustrate the concepts discussed as applicable to a general research lab. This lack of applied concepts also led to confusion regarding the underlying EQUIPD initiative itself: there was confusion regarding its implementation as software vs advisory committee vs regulatory philosophy. In addition to providing examples of how to apply the EQUIPD principles in a practical, concrete example, one reviewer suggested that working with journals to help them understand and check submissions for compliance with such a set of rules may be a worthwhile endeavor; in other words, EQUIPD would seem meaningless if compliance checkers did not exist as part of the overall process. Lastly, a revision of the introduction and context of the work is recommended; reviewers took issue with the notion that reduced drug approvals over recent years is attributable to poor preclinical rigor.

---

## [Author Response]

Essential revisions:The reviewers noted that the paper was noble for taking on the topic of false or misleading research findings and their goal to improve research rigor and honesty. Moreover, their introduction of a host of processes and box-checking steps to incorporate into research would be appreciated by lab managers. However, the reviewers note that, written in its current form as a process document, there were few concrete examples provided to illustrate the concepts discussed as applicable to a general research lab.

We have added a new Table 3 to illustrate “what this is all about” – i.e. five basic principles that explain how the EQIPD QS operates. For each principle, we use an example of randomization for a specific illustration how principles are applied.

This lack of applied concepts also led to confusion regarding the underlying EQUIPD initiative itself: there was confusion regarding its implementation as software vs advisory committee vs regulatory philosophy.

We have removed information about the tools and have revised the text to make it clear that this manuscript presents a novel Quality System rather than software or anything else. We have also provided additional illustrations such as Figure 2 that illustrates optional use of tools developed by EQIPD. Finally, we have added a new Table 5 that provides an overview of how the EQIPD framework cane be used.

In addition to providing examples of how to apply the EQUIPD principles in a practical, concrete example, one reviewer suggested that working with journals to help them understand and check submissions for compliance with such a set of rules may be a worthwhile endeavor; in other words, EQUIPD would seem meaningless if compliance checkers did not exist as part of the overall process.

We have added a new section describing the assessment processes that have been developed by EQIPD.

Lastly, a revision of the introduction and context of the work is recommended; reviewers took issue with the notion that reduced drug approvals over recent years is attributable to poor preclinical rigor.

We have revised the text to make sure that reduced drug approvals are attributable to lacking research rigor. In fact, we emphasize that research rigor is merely “one area requiring immediate attention”.